# Identifying Patients with Bicuspid Aortic Valve Disease in UK Primary Care: A Case–Control Study and Prediction Model

**DOI:** 10.3390/jpm12081290

**Published:** 2022-08-05

**Authors:** William Evans, Ralph Kwame Akyea, Stephen Weng, Joe Kai, Nadeem Qureshi

**Affiliations:** 1Primary Care Stratified Medicine (PRISM), Centre for Academic Primary Care, School of Medicine, University of Nottingham, Nottingham NG7 2RD, UK; 2Statistical Decision Sciences, Cardiovascular and Metabolism, Janssen Research and Development, High Wycombe HP12 4EG, UK

**Keywords:** electronic medical records, primary care, aortic valve, prediction model

## Abstract

Bicuspid aortic valve disease (BAV) is the most common congenital heart condition, and early detection can improve outcomes for patients. In this case–control study, patients with a diagnosis of BAV were identified from their electronic primary-care records in the UK’s Clinical Practice Research Datalink (CPRD). Each case was propensity-score matched to up to five controls. The clinical features recorded before diagnosis were compared. The proposed clinical features shown to be associated with BAV (*p* < 0.05) were incorporated into a multivariable regression model. We identified 2898 cases. The prevalence of BAV in the CPRD was 1 in 5181, significantly lower than expected, suggesting that diagnosis and/or recording could be improved. The following biologically plausible clinical features were associated with a subsequent diagnosis of BAV: palpitations (OR: 2.86 (95% CI: 1.60, 3.16)), atrial fibrillation (AF) (OR: 2.25 (95% CI: 1.60, 3.16)) and hypertension (OR: 1.72 (1.48, 2.00)). The best model had an AUC of 0.669 (95% CI: 0.658 to 0.680), a positive predictive value (PPV) of 5.9% (95% CI: 4.0% to 8.7%) and a negative predictive value (NPV) of 99% (95% CI: 99% to 99%) at a population prevalence of 1%. This study indicates that palpitations, hypertension and AF should trigger a clinical suspicion of BAV and assessment via echocardiography. It also demonstrates the potential to develop a prediction model for BAV to stratify patients for echocardiography screening.

## 1. Introduction

Bicuspid aortic valve (BAV) disease is the most common congenital heart condition, present in 0.5–2% of the general population, and is responsible for more complications and deaths than all other congenital heart conditions combined [1,2]. BAV is a highly heterogeneous valvulo-aortopathy with significant variation in its aortic and valvular phenotypes, associated disorders, complications and prognosis [3,4]. Early recognition of BAV is associated with improved outcomes, enabling suitable surveillance, lifestyle advice, cardiovascular risk management and timely surgery [5,6].

The presentation and consequences of BAV are varied but can be subdivided into three valvulo-aortopathy presentations. The first group is a more complex presentation, with BAV often part of a broader syndrome presenting in childhood or adolescence with accelerated disease. The other two presentations constitute most cases, in which the valvulo-aortopathy is non-syndromic and is an isolated and sporadic clinical feature in an otherwise healthy individual. These include an uncomplicated or undiagnosed presentation that does not manifest clinically and is usually identified incidentally, and a more common ‘typical’ presentation with progressive valvular dysfunction and aortic dilatation. The Olmsted County community study followed 212 patients with no cardiovascular symptoms and no or minimal valvular dysfunction at diagnosis for a further 25 years. Aortic stenosis (AS) requiring aortic valve replacement (AVR) occurred in 50% of patients, and significant aortic dilatation occurred in more than 25% of patients [7]. Unfortunately, there remain few ways to predict who will and who will not progress, or their rate or likelihood of complications [4]. The earlier identification of the most common ‘typical’ BAV presentation in patients, therefore, presents a major clinical challenge. Many are only diagnosed when complications and cardiac damage have occurred, but early identification can enable long-term surveillance, timely surgical intervention and the avoidance of complications [3].

The route to diagnosis typically begins in primary care, with a referral for transthoracic echocardiography (TTE) following the development of symptomatic complications. For those who were asymptomatic at diagnosis, this may follow the identification of a murmur during auscultation as an incidental finding when cardiac imaging is performed for another reason, or when screened for BAV, in those with a known affected first-degree relative [7,8]. 

Early identification is key to improving outcomes, and there is a need for a greater understanding of its early clinical features and the path to diagnosis [5], especially in primary care, where the path to diagnosis begins. In this study, we explore the clinical features that precede the diagnosis of BAV in UK primary care and use these to develop a predictive model that may assist in earlier case findings. 

## 2. Materials and Methods

### 2.1. Data Source

The UK Clinical Practice Research Datalink (CPRD), a database of electronic medical records, holds longitudinal data from 1987 to the present. It has 681 primary-care practices’ data, encompassing 35 million patients’ lives, with 15 million currently registered [9]. 

This study was approved by the Independent Scientific Advisory Committee for the Medicines and Healthcare Products Regulatory Agency database research (ISAC Protocol 19_049). The reporting of this study follows the TRIPOD guidelines [10].

### 2.2. Study Design and Population

All individuals with a BAV diagnostic code (CTV3 Read code: Bicuspid Aortic Valve (P641.00)) were retrieved from the CPRD. The index date for diagnosis was taken as the date the code first appeared in the patient’s record. 

Patients were eligible if registered with their practice for at least 12 months, and data were collected from the time their practice’s data were deemed to meet the CPRD’s standard until the date of data extraction in July 2018. 

Propensity-score matching, using sex, age, body mass index (BMI), smoking status and ethnicity (White or non-White), was used to match up to 5 controls from the same practice as each case. Propensity-score matching allows observational studies to minimise bias by balancing the observed baseline covariates between groups [11]. 

Following a review of the literature and discussions with a range of clinicians and experts, features that may occur in advance of a BAV diagnosis were identified. These features were mapped to the appropriate electronic health record (EHR) codes and the data set searched to identify those appearing in advance of the diagnostic date for both the BAV cases and the equivalent age of each of the cases’ 5 matched controls. To estimate missing values for blood pressure, pulse rate and BMI, multiple imputation with chained equations was used to generate 10 imputed data sets and was pooled using Rubin’s rules [12] (Appendix A).

### 2.3. Statistical Analysis and Model Development

Clinical features, based on clinical review and biological plausibility, were evaluated for their significance as potential predictors of later BAV diagnosis via univariate logistic regression. Those found to be significantly associated (*p* value < 0.05) were incorporated into a multivariable logistic-regression model. Variations of model composition based on different clinical features were compared using model-fit parameters, favouring models with lower Akaike information criterion (AIC) and Bayesian information criterion (BIC) to prevent over-fitting. Model performance was evaluated using AUC and calibration [13]. Further iterations of the model were performed using both logarithmic transformations of continuous variables and the calculation of the fractional polynomials of these same variables to improve model calibration. The optimal probability cut-off was calculated using several methods: the maximum-product index between sensitivity and specificity [14], the Youden index [15] and an approach based on PPV and NPV. 

Validation was performed via bootstrapping, as described by Harrell et al. (1996) [16]. This involved repeatedly resampling the data set 200 times for each sample of varying size. The model was then fitted to each of these 200 data sets, with each fitted model then applied to the resampled data. The mean AUC from the refitted model for each of these 200 data sets was then calculated, and the difference between this and the AUC of the original data set’s model was calculated. The original AUC minus this difference was then calculated to give an optimism adjusted AUC. 

In a subsequent sensitivity analysis, to exclude those diagnosed with BAV as an incidental finding at an echocardiography examination performed due to another indication, we re-analysed the model in a sub-population that excluded patients who had any of a series of diagnostic codes (tachycardia, bradycardia, AF, hypertension, heart failure, coronary artery disease, stroke, collapse, endocarditis, other valvular disease, aortic dissection or palpitations) recorded in the 90 days before their BAV diagnosis. Ninety days was chosen, as this was felt to capture the period of time taken to allow a referral from primary care for echocardiography, for testing to be performed, and for the findings to be captured in the EHR [17].

All analyses were performed using Stata 15.1 (StataCorpLP, College Station, TX, USA).

Patients were involved in developing the initial concept of this study but were not involved in its design, conduct, reporting or dissemination.

## 3. Results

### 3.1. Baseline Characteristics

There were a total of 17,385 individuals in this study, including 2898 patients with BAV and 14,487 controls.

The period prevalence in this population (for the duration of data capture in the CPRD) was 193 per million (0.0002%). 

In our sample, most patients with BAV were male (61.84%). They were predominantly either a normal weight or overweight (BMI >/= 18.5 to 30) (85%), with a median IQR of 24.8 (22.6–27.3) kg/m^2^. Most were non-smokers (62%). Ethnicity was poorly recorded; most (55%) had no record of ethnicity, and in those for whom it was recorded, they were almost entirely White (96%). The age at the diagnosis of BAV had a mean (SD) of 41.27 (16.6) years and a median (IQR) of 39.17 (27.26, 53.41). The controls had similar demographic features (Table 1).

Most cases had no BAV complications or cardiovascular disease before the BAV diagnosis (Table 2). 

### 3.2. Associated Features and Multivariate Modelling

Certain clinical features were strongly associated with a subsequent diagnosis of BAV, including hypertension, dizziness, atrial fibrillation and palpitations (Table 3), with most features recorded more than 90 days before BAV diagnosis. The median time in days was as follows: hypertension, 1232; dizziness, 1420; AF, 419; and palpitations, 655 (Table 4). 

The optimum model, incorporating these clinical features and others, including the logarithmic transformation of mean pulse (Table 3), had an AUC of 0.669 (95% CI: 0.658 to 0.680). The overall calibration slope of the model was 1.0 (Figure 1). Previous iterations of the model during its development can be found in Appendix A. The model’s performance is demonstrated in Figure 2, with positive and negative predictive values at different population prevalence levels in Table 5. The optimum trade-off of sensitivity and specificity for both the Youden Index and the maximal product index occurred at a 15% probability (sensitivity 61.0% and specificity 64.8%). However, given that BAV is rare, setting a probability cut-off based on optimal PPV and NPV makes more clinical sense. Using this approach at a 65% probability at a population prevalence of 1% gave PPV = 5.9% and NPV = 99% (Table 5).

The bootstrap analysis with 200 replication samples generated a mean AUC of 0.667 (95% CI: 0.656, 0.678). The difference between this and the original data set’s AUC was utilised to calculate an optimism adjusted AUC of 0.6701 (95% CI: 0.660, 0.684).

## 4. Discussion

### 4.1. Principal Findings

This large population case–control study is the first to explore the presentation of patients with BAV in primary care and to analyse what is captured in their clinical records. We have found a range of clinical features coded in their EHRs that precede a BAV diagnosis, including hypertension, AF, palpitations, stroke/TIA, dizziness and collapse. Atrial fibrillation and palpitations were particularly strongly associated with BAV, and their presence should thus raise the suspicion of aortic valve disease and lower a primary-care clinician’s threshold for echocardiography. This test is currently not routinely performed following such presentations in UK primary care.

While not designed as a prevalence study, the period prevalence in this population was 0.0002%, well below the estimated population prevalence of 0.5–2.0% [2]. This suggests that BAV is underdiagnosed and/or under-recorded in UK primary care. By combining seven clinical features, we have developed a new risk-prediction model to stratify patients’ risks of developing BAV, with an AUC of 0.669.

### 4.2. Comparison with Other Literature/Studies

This is the first available piece of research to capture the clinical features in primary care that precede a BAV diagnosis. Observational studies of BAV patients, registry data and cohort studies often include limited information on patients’ symptoms, mainly reporting the direct complications of the valve and aorta, the need for interventions, and mortality rates [7,18]. Patients in these studies are either those identified incidentally or through cascade testing and are presumed to be asymptomatic, or for those patients with the complications of BAV, their symptoms were inferred from what would be typical of each complication (e.g., congestive heart failure, breathlessness, fatigue, aortic stenosis, palpitations, dizziness and breathlessness) [18]. In contrast, in the current case–control study, the features recorded in their EHRs before diagnosis were examined. 

In our study, certain clinical features were identified that occurred at a significantly higher frequency in advance of a BAV diagnosis compared to the propensity-matched control group. 

Palpitations occurred at an increased frequency (6.6% vs. 1.8%) in patients documented to have had this symptom 21 months before diagnosis. 

AF occurred at a significantly increased frequency (2.8% vs. 0.59%) and was typically recorded 14 months before a BAV diagnosis. AF often coexists with AS and is known to occur both earlier and more frequently in BAV [19]. 

Other non-specific symptoms included dizziness, recorded in 6.5% of cases and 3.7% of controls, and collapse, recorded in 4.0% vs. 2.2%. The potential aetiology of both of these diagnoses is broad, but includes AR, present in approximately 50% of BAV patients [18] and AS [20]. 

A diagnosis of hypertension was significantly more common (11.6% vs. 5.0%). Hypertension is known to exacerbate BAV, both increasing symptoms and worsening outcomes for the same degree of valvular dysfunction [21], and in the general population, raised blood pressure is associated with an increased risk of both AS and AR [22].

Coronary artery disease (CAD) diagnoses were more common in those with BAV (3.7% vs. 1.6%). Echocardiography was more likely to be performed in those with CAD; however, 73% of patients with CAD and BAV received their CAD diagnosis > 90 days before their BAV diagnosis, suggesting that most patients’ BAV diagnoses did not occur because of their CAD diagnosis. Although a link between BAV and CAD is not clear, BAV is associated with a 2–4-fold increased rate of a left-dominant coronary artery system, an anatomical variant associated with increased rates of CAD [23].

Most patients in this study had no recorded aortic valvular disease complications before diagnosis (Table 2). This differs from the Olmsted community study, in which 60% had moderate or severe valve disease at diagnosis [7]. 

### 4.3. Clinical Implications

An early diagnosis of BAV allows for the optimisation of management and follow-up, avoiding complications and improving outcomes. This study identifies key clinical features that are recorded in BAV patients’ primary-care records before diagnosis (OR): AF (2.25), palpitations (2.86) and hypertension (1.72). Given that BAV has a relatively common prevalence of 0.5–2% [2], we propose that these features alone should raise clinical suspicion of this diagnosis, and should prompt assessment using echocardiography (TTE). 

Patients diagnosed with AF often receive a TTE as part of their clinical investigation, particularly if there is a suspicion of underlying structural heart disease. However, the English national guidelines (NICE, 2021) do not recommend TTE for all [24]. This differs from the European Society of Cardiology guidelines, in which TTE is recommended for all newly diagnosed patients [25]. Currently, only a proportion of patients presenting with palpitations or transient loss of consciousness will undergo TTE as part of their workup (e.g., when a murmur is identified and structural disease suspected) [26,27]. Similarly, in hypertensive patients, TTE is often performed to assess for evidence of end-organ damage; however, it is not routinely performed, nor recommended, in England [28]. 

In addition to the identification of aortic valve disease, an investigation with echocardiography may have added benefits for these patients, identifying other clinical features and enabling stratification for subsequent therapies. 

BAV has high heritability; the risk in first-degree relatives is high (10 to 30%). Index case identification enables echocardiography screening of family members [8]. 

In this study, these strongly associated clinical features were combined into a multivariate model, enabling greater refinement of BAV patient identification, with discrimination reflected by an AUC of 0.669 and a maximum PPV of 3%, 5.9% and 11.3% at a prevalence of 0.5%, 1% and 2%, respectively. This compares favourably with current suspected-cancer referral guidance, which uses a positive predictive value threshold of 3% for clinical features to warrant further investigation [29].

Screening for valvular disease has already been proposed in patients over the age of 75 who have additional risk factors [6]. Advances in handheld echocardiograms enable more rapid and cost-effective community screening. This suggests that the community screening of those at risk of valvular disease is an increasingly realistic proposition for practice [6,30].

The prevalence of BAV in this population was significantly lower than expected, likely reflecting both underdiagnosis and under-recording in primary care. Addressing these omissions is important. Patients who are diagnosed but not suitably recorded cannot be easily identified, prescription decisions cannot be considered in light of the diagnosis and patients cannot be easily recalled for surveillance, cascade testing or audit purposes.

### 4.4. Strength and Limitations

This case–control study was derived from a high-quality primary-care database, broadly representative of the general population of the UK and reflective of the real-world experiences of UK patients. The use of propensity-score matching has minimised bias by balancing covariates between the cases and the controls. The model was based on clinical variables routinely collected in primary care as part of standard care. Most identified features occurred more than 90 days before the BAV diagnosis, suggesting that these were not directly linked to the referral for the diagnostic test that identified BAV. While the miscoding of medical conditions is an acknowledged limitation of electronic health records, for BAV, there is no clearly ambiguous coding nomenclature for another similar disease. Moreover, although we were unable to ascertain how the diagnosis was made, in the vast majority, this will follow a TTE, which has high specificity for BAV (88.3–97.2%) [31]. 

The multivariable model was internally validated via bootstrapping, which demonstrated that the model performed similarly across the 200 repetitions. 

The limitations of this study included ascertainment bias. The BAV cases may have had greater clinical involvement in advance of their diagnosis, with the greater recording of coded entries a reflection of clinical contact, rather than a difference in frequency. The prevalence of BAV in this population was far lower than the expected prevalence of 0.5–2% [2]; consequently, a proportion (similar to this expected population prevalence) of undiagnosed or uncoded BAV patients will sit within the control group. 

Additionally, the limitations of information bias, bias due to missing data and residual or unmeasured confounding are acknowledged; these limitations are shared with other database and population studies. 

### 4.5. Further Research

The findings of this study demonstrate the potential of developing a risk-prediction tool for BAV to target screening with echocardiography. Many of the risk factors in the model would also be expected to be present in valvular diseases of other aetiologies, all of which would then require echocardiography. Combining these valvular diseases to develop a single predictive tool may enable more efficient case finding. Although internally validated, before implementation, the current risk-prediction tool should be validated in other primary-care datasets. 

## Figures and Tables

**Figure 1 jpm-12-01290-f001:**
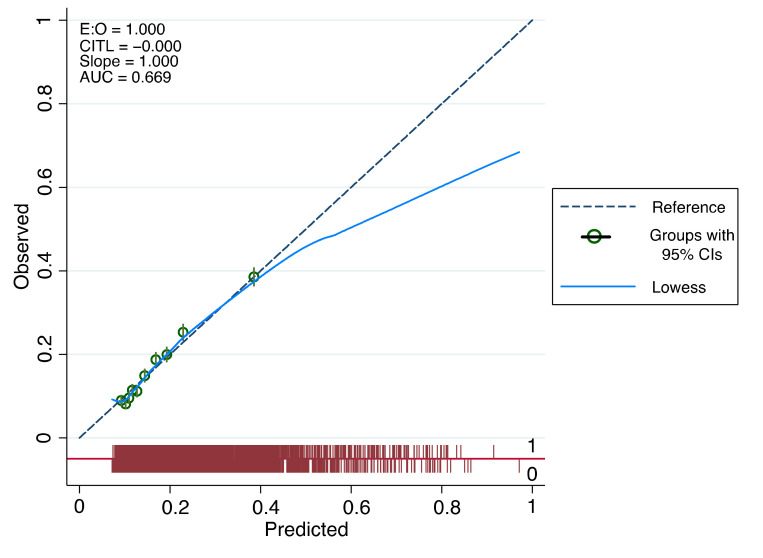
Assessing calibration of model.

**Figure 2 jpm-12-01290-f002:**
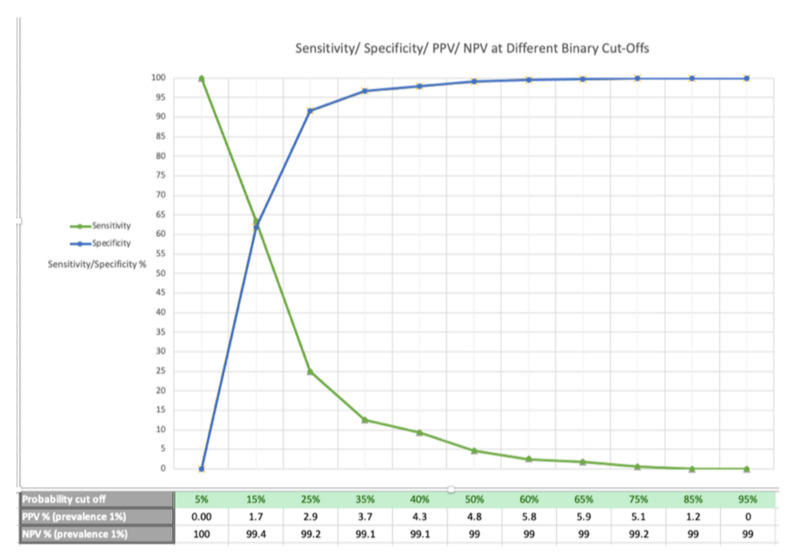
Sensitivity/specificity/positive predictive values (PPV)/negative predictive values (NPV) for a prevalence of 1% at different probability cut-offs.

**Table 1 jpm-12-01290-t001:** Characteristics of case and controls by gender.

		Cases BAV n = 2898 (16.67)		Controls n = 14,487 (83.33)	
		Male	Female	Male	Female
	n (%)	1792 (61.84)	1106 (38.16)	8811 (60.82)	5676 (39.18)
Age at diagnosis *	Mean (s.d)	41.28 (16.46)	41.82 (17.11)	41.28 (16.46)	41.38 (16.86)
	Median	39.67	38.85	39.67	38.24
	Range	16.02, 90.84	16.00, 90.37	16.02, 90.84	16.02, 90.4
	IQR	27.19, 53.88	27.51, 53.78	27.19, 53.88	27.27, 53.01
BMI (kg/m^2^)	Median (IQR)	25.35 (22.7, 27.9)	24.2 (21.8, 27.5)	25 (22.6, 28.1)	24 (21.4, 27.4)
Smoking status	Smoker n (%)	371 (20.70)	174 (15.73)	1892 (21.47)	919 (16.19)
	Non-smoker n (%)	1045 (58.31)	750 (67.81)	3751 (42.57)	3008 (53.00)
	Ex-smoker n (%)	155 (8.65)	86 (7.78)	636 (7.22)	428 (7.54)
	No data n (%)	221 (12.33)	96 (8.68)	2532 (28.74)	1321 (23.27)
Ethnicity	White n (%)	777 (43.36)	469 (42.41)	2457 (27.89)	1601 (28.21)
	Non-White/unknown n (%)	1015 (56.64)	637 (57.59)	6354 (72.11)	4075 (71.79)

(* For controls, diagnostic date is taken as the date for the matched case).

**Table 2 jpm-12-01290-t002:** Frequency of BAV complications or CV disease before diagnosis by age.

Age Categories	Number	No Preceding Complications * n (%)	No Preceding Complications or CVD^+^ n (%)	CAD Dx before n (%)	CHF Dx before n (%)	Endocarditis Dx before n (%)	Stroke Dx before n (%)
<25	566	526 (93%)	517 (91%)	1 (0.18)	1 (0.18)	1 (0.18)	0
25–49	1463	1271 (87%)	1189 (81%)	12 (0.82)	8 (0.55)	11 (0.75)	9 (0.62)
50–74	779	582 (75%)	423 (54%)	80 (10.27)	15 (1.93)	4 (0.51)	17 (2.18)
>=75	90	57 (63%)	42 (47%)	14 (15.56)	7 (7.78)	0	2 (2.22)

(* No valvular disease, CHF, CAD, aortic dissection or aneurysm, or endocarditis. ^+^ As per * and no AF, peripheral arterial disease, hypertension or stroke.).

**Table 3 jpm-12-01290-t003:** Clinical features incorporated into final multivariable analysis.

Clinical Variable	Odds Ratio [95% CI]	*p*-Value	Coefficient [95% CI]
Diagnosis of hypertension	1.72 [1.48 to 2.00]	0.000	0.543 [0.391 to 0.695]
Diagnosis of atrial fibrillation [AF]	2.25 [1.60 to 3.16]	0.000	0.810 [0.470 to 1.15]
Diagnosis of palpitations	2.86 [2.32 to 3.51]	0.000	1.05. [0.843 to 1.26]
Diagnosis of dizziness	1.22 [1.01 to 1.47]	0.039	0.198 [0.010 to 0.386]
Ethnicity White/non-White	1.35 [1.29 to 1.41]	0.000	0.301 [0.259 to 0.343]
Log mean pulse^3,3^		0.000	−1.38 [−1.98 to −0.788] 0.742 [0.410 to 1.07]
Beta-blocker category			
Intermittent beta-blocker use (Longest gap between prescriptions > 90 but <180 days)	2.05 [1.75 to 2.41]	0.000	0.717 [0.558 to 0.878]
Intermittent beta-blocker use (Longest gap between prescriptions > 180 days)	2.23 [1.79 to 2.79]	0.000	0.804 [0.581 to 1.03]
Continuous beta-blocker use (No gap between prescriptions longer than 90 days)	1.71 [0.927 to 3.17]	0.086	0.539 [−0.759 to 1.15]

**Table 4 jpm-12-01290-t004:** List of associated features and time before diagnosis.

Diagnoses before BAV Diagnosis n (%) (Unless Otherwise Stated)	Cases	Controls	*p* Value (chi^2^) (* = *t* Test)	Time before Diagnosis (Days) Median [IQR]	First Dx > 90 Days before BAV Dx (%)
	n = 2898	n = 14,487			
Cardiovascular					
Systolic BP (mean [SD])	127.4 [18.6]	125.7 [18.6]	<0.001 *		
Diastolic BP (mean [SD])	76.3 [11.0]	76.2 [11.43]	0.7 *		
Diagnosis of hypertension	337 [11.6]	727 [5.02]	<0.001	1232 [351–2547]	85
Pulse (mean [SD])	74.9 [14.85]	76.83 [13.37]	0.03 *		
Diagnosis of tachycardia	34 [1.17]	44 [0.30]	<0.001	1147 [34–5423]	74
Diagnosis of bradycardia	10 [0.35]	22 [0.15]	0.027	1830 [172–2475]	80
Tachycardia on pulse (mean >100)	22 [0.76]	52 [0.36]	<0.001		
Bradycardia on pulse (mean < 60)	56 [1.93]	70 [0.48]	<0.001		
Diagnosis of aortic aneurysm/dissection	18 [0.62]	11 [0.08]	<0.001	265 [47–1274]	72
Diagnosis of endocarditis	16 [0.55]	1 [0.01]	<0.001	954 [193–3893]	88
Diagnosis of tricuspid valve disease	6 [0.21]	7 [0.05]	0.004	1341 [1113–2749]	100
Diagnosis of mitral valve disease	61 [2.10]	18 [0.12]	<0.001	398 [132–1431]	82
Diagnosis of pulmonary valve disease	18 [0.62]	9 [0.06]	<0.001	1325 [255–1571]	94
Diagnosis of palpitations	192 [6.63]	258 [1.78]	<0.001	655 [163–1756]	81
Diagnosis of heart failure	31 [1.07]	52 [0.36]	<0.001	246 [58–946]	65
Diagnosis of coronary arterial disease	107 [3.69]	231 [1.59]	<0.001	731 [85–1866]	73
Diagnosis of atrial fibrillation (AF)	81 [2.80]	86 [0.59]	<0.001	419 [108–1187]	79
Neurological					
Diagnosis of stroke/TIA	28 [0.97]	72 [0.50]	0.002	888 [142–4899]	93
Diagnosis of epilepsy	29 [1.00]	95 [0.66]	0.044	839 [186–2701]	83
Diagnosis of migraine	27 [0.93]	63 [0.43]	<0.001	1015 [290–3005]	96
Diagnosis of dizziness	187 [6.45]	538 [3.71]	<0.001	1420 [374–2943]	90
Diagnosis of collapse	116 [4.00]	312 [2.15]	<0.001	855 [152–2304]	83
Beta-blocker use					
Not taking	2427 [83.75]	13,649 [94.2]			
Intermittent use (90–180-day gap in prescription issue after initiation)	306 [10.56]	558 [3.85]			
Intermittent use (>180-day gap in prescription issue after initiation)	148 [5.11]	247 [1.70]			
Continuous use (<90-day gap in prescription issue since initiation)	17 [0.59]	33 [0.22]	<0.001		

**Table 5 jpm-12-01290-t005:** Positive predictive value (PPV) and negative predictive value (NPV) of the model at population prevalence 0.5%, 1% and 2%.

Probability Cut-Off	5%	15%	25%	35%	50%	65%	75%	85%	95%
PPV % (prevalence 0.5%)	0	0.8	1.5	1.9	2.4	3	2.6	0.6	0
NPV % (prevalence 0.5%)	100	99.7	99.6	99.5	99.5	99.5	99.5	99.5	99.5
PPV % (prevalence 1%)	0	1.7	2.9	3.7	4.8	5.9	5.1	1.2	0
NPV % (prevalence 1%)	100	99.4	99.2	99.1	99.0	99.0	99.2	99.0	99.0
PPV % (prevalence 2%)	0	3.3	5.7	7.2	9.2	11.3	9.7	2.5	0
NPV % (prevalence 2%)	100	98.8	98.4	98.2	98.1	98.0	98.0	98.0	98.0

## Data Availability

Data supporting these results are available from the CPRD. Code lists used to perform the analysis are available on request.

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
