# Peer review of "Identifying Patients with Bicuspid Aortic Valve Disease in UK Primary Care: A Case–Control Study and Prediction Model"

_jpm, 2022, doi:10.3390/jpm12081290_

Round 1

Reviewer 1 Report

Clearly written and well presented data. Complex statistical analysis that should be reviewed by qualified statistician. Interesting conclusions that may well find way to routine clinical practice in primary care. No further comments.

Reviewer 2 Report

Per usual, the diagnosis of Bicuspid Aortic Valves (BAV) before the development of serious complications is an accidental finding during echocardiographic exam. Therefore, the purpose of this article is to develop a risk prediction tool for BAV to target screening with echocardiography.

This study indicates that palpitations, hypertension, and AF should trigger clinical suspicion of BAV and echocardiography.  However, these clinical features seem to be nonspecific for BAV and common for many heart diseases, not only “valvular disease of other aetiologies” (line 306). As a result, echocardiography should be performed to assess any underlying structural heart disease in these patients and this risk prediction model does not seem entirely accurate and specific to BAV. It may not easily function in clinical practice as a tool for echocardiographic screening. However, an attempt to create such a model is useful for early diagnosis of this common heart defect to avoid complications and improve outcomes.

Finally, this model is based on reliable clinical data and on a suitable statistical tool. The paper is written competently and consistently and may be recommended for publication.